# Lung colonization by *Aspergillus fumigatus* is controlled by ZNF77

Sara Gago [1], Nicola L.D. Overton[1,5], Nagwa Ben-Ghazzi[1], Lilyann Novak-Frazer [2,3], Nick D. Read[1], David W. Denning[4] & Paul Bowyer[1]

*Aspergillus fumigatus* is a critical pathogen of humans. Exposure to *A. fumigatus* conidia occurs frequently but is normally cleared from the respiratory airways. In contrast, individuals with respiratory diseases are often highly colonized by fungi. Here, we use genome-edited epithelial cells to show that the genetic variant rs35699176 in ZNF77 causes loss of integrity of the bronchial epithelium and increases levels of extracellular matrix proteins. These changes promote *A. fumigatus* conidial adhesion, germination and growth. RNA-seq and LC/MS-MS analysis reveal rs35699176 upregulates vesicle trafficking leading to an increment of adhesion proteins. These changes make cells carrying rs35699176 more receptive to *A. fumigatus* in the early stages of infection. Moreover, patients with fungal asthma carrying rs35699176 $^{+/-}$ have higher *A. fumigatus* loads in their respiratory airway. Our results indicate ZNF77 as a key controller of *Aspergillus* colonization and suggest its utility as a risk-marker for patient stratification.

[1] Manchester Fungal Infection Group, Division of Infection, Immunity and Respiratory Medicine, University of Manchester, CTF Building, 46 Grafton Street, Manchester M13 9NT, UK. [2] Division of Infection, Immunity and Respiratory Medicine, School of Biological Sciences, Manchester Academic Health Science Centre, The University of Manchester and University Hospital of South Manchester NHS Foundation Trust, Manchester M23 9LT, UK. [3] Mycology Reference Centre, ECMM Excellence Centre of Medical Mycology, Manchester University NHS Foundation Trust, Manchester M23 9LT, UK. [4] National Aspergillosis Centre, Manchester Academic Health Science Centre, University Hospital of South Manchester NHS Foundation Trust, Manchester M23 9LT, UK. [5] Present address: Clinical & Experimental Pharmacology Group, CRUK Manchester Institute, University of Manchester, Manchester M20 4GJ, UK. Correspondence and requests for materials should be addressed to P.B. (email: paul.bowyer@manchester.ac.uk)

Respiratory exposure to airborne *Aspergillus fumigatus* conidia is universal and unescapable[1]. In the healthy host, fungal spores are rapidly cleared from the respiratory airways. However in patients with an immune defect such as neutropenia, asthma or a cavitating lung disease, *A. fumigatus* spores can persist, colonize and lead to the development of aspergillosis[2]. Aspergillosis has recently been shown to cause more than 400,000 deaths each year[3].

*A. fumigatus* colonization of the respiratory is associated with a higher risk of invasive aspergillosis and death in patients undergoing lung transplantation[4,5] and high IgG levels in patients with chronic pulmonary aspergillosis[6]. Moreover, increased loads of *A. fumigatus* in the respiratory airways of asthmatic patients is associated with the development of allergic bronchopulmonary aspergillosis (ABPA), a progressive fungal allergic lung disease that significantly reduces the quality of life of over 5 million asthmatic people worldwide[7–10]. *Aspergillus* colonization of the lung epithelium in patients with ABPA leads to a hypersensitivity reaction to fungal antigens that promotes an IgE-mediated eosinophilic response, high mucus secretion and airway obstruction[10]. Fungal burden in ABPA varies considerably from individual to individual and colonization is associated with airway disease severity[11]. Abnormalities in the airway mucosal defences of patients with asthma and cystic fibrosis[12], corticosteroid treatment[13] or antibiotic misuse[14] have been described as risk factors for *Aspergillus* allergic reaction and colonization in ABPA. However, fungal allergy only affects a low percentage of asthmatics in spite of constant exposure suggesting that ABPA might be due to an impaired genetic capability of some asthmatic patients to fend off fungal colonization[15].

Currently, only a few polymorphisms in genes predicted to play a crucial role in the immune response to *A. fumigatus* have been assessed for their association with ABPA[16–20]. These genes all play a role in the adaptive immune response to fungi and are assumed to exacerbate fungal allergy in the context of atopic asthma.

The transcription factor ZNF77 belongs to the zinc finger protein family. Bioinformatics modelling suggests this transcription factor to control defensins, elastase and calmodulin expression, all potentially important for fungal clearance by the lung epithelium[21] (see supplementary note 1 for additional information). Here, we describe the role of rs35699176 in controlling fungal colonization in ABPA. This variant introduces a premature stop codon before the DNA binding region in the transcription factor ZNF77. Based on these results, we propose that ZNF77-genotyping of patients with ABPA may be a useful risk-marker for fungal colonization.

## Results

**rs35699176 impairs epithelial integrity**. To investigate the role of rs35699176 in fungal colonization, we genome-edited 16HBE bronchial epithelial cells to carry the rs35699176 variant using CRISPR/Cas9. The insertion of the mutation in the 16HBE genome was determined by PCR (Fig. 1a) indicating that the CRISPR/Cas9 system correctly introduces the specific mutation into 16HBE cells. ZNF77 gene expression was not significantly different ($P = 0.94$) in parental 16HBE and the genome-edited cell line (Fig. 1b).

We first investigated the role of rs35699176 in controlling epithelial monolayer integrity. 16HBE and 16HBE$^{rs35699176}$ cell lines were seeded in transwell inserts and allowed to form monolayers over 9 days. We observed an 87% reduction of confluence in 16HBE$^{rs35699176}$ ($162.6 \pm 21.60$ Ohms per cm$^2$) compared to 16HBE cells ($917.5 \pm 103.138$ Ohms per cm$^2$) after 9 days ($P < 0.001$) (Fig. 1b). To investigate morphological changes

associated with this loss of confluence, epithelial monolayers were visualised using live-cell confocal microscopy. Imaging of 16HBE$^{rs3569917}$ epithelial monolayers demonstrated defective adhesion to substrate and loss of confluence (Fig. 1c, Supplementary Movies 1 and 2). To examine if this loss of confluence in 16HBE$^{rs35699176}$ affects the bronchial epithelial interaction with *A. fumigatus*, we initially screened expression of genes (occludin, caveoline and E-cadherin) known to be involved in tight junction and cell adhesion component regulation. Surprisingly we observed that occludin expression doubled in 16HBE$^{rs35699176}$ cells compared to 16HBE. Occludin expression decreases rapidly after *A. fumigatus* exposure ($P < 0.005$). No changes in expression were detected in the other genes (Fig. 1d).

**rs35699176 promotes *A. fumigatus* adhesion**. Human lung epithelium employs several methods of defence to remove and destroy inhaled pathogens. Besides physical removal by mucociliary flow, epithelial cells can secrete a wide range of antimicrobial peptides and they can also act as non-professional phagocytes[22]. Moreover, they secrete chemotactic factors to recruit more specialized immune cells to contribute to fungal clearance[23]. Most clearance mechanisms mediated by epithelium and macrophages are effective against ungerminated conidia. The capability of *A. fumigatus* to more rapidly adhere, germinate and grow on the bronchial epithelium could lead to increased survival of the host immune response facilitating fungal colonization[24].

Adhesion of *A. fumigatus* spores was 10.4% higher on 16HBE$^{rs35699176}$ cells compared to 16HBE at 2 h post-infection ($P < 0.001$) (Fig. 2a). Similarly, adherence to extracellular matrix was 35% higher in 16HBE$^{rs35699176}$ than 16HBE cells ($P < 0.0001$) and this was associated with higher concentration of extracellular matrix proteins in 16HBE$^{rs35699176}$ cells ($P < 0.00001$) (Fig. 2b, c).

The faster adhesion time observed for *A. fumigatus* conidia on 16HBE$^{rs35699176}$ cells led to earlier *A. fumigatus* germination in our model. In order to rule out possible isolate-specific effects, six *A. fumigatus* isolates were tested for germination and adherence to the modified cell line. All six *A. fumigatus* strains tested were able to germinate $28.3 \pm 12.6$ min earlier in the presence of 16HBE$^{rs35699176}$ cells than in the presence of 16HBE cells (range 15–41 min) ($P < 0.001$) (Supplementary movies 3 and 4) and to double overall hyphal extension at 9 h post-inoculation ($76.4 \pm 5.0$ μm in the presence of 16HBE cells compared to $157.7 \pm 10.7$ μm in the presence of 16HBE$^{rs35699176}$cells, $P < 0.001$) (Fig. 2c, d and Supplementary Figure 1). These growth changes in *A. fumigatus* led to a 6.51% increment of *A. fumigatus*-mediated cytotoxicity in the 16HBE$^{rs35699176}$ cells ($P < 0.05$) determined by a lactate dehydrogenase assay at 16 h post-inoculation (Fig. 2e). Three other closely related pathogenic *Aspergillus* species, *Aspergillus nidulans*, *Aspergillus terreus* and *Aspergillus niger*, did not show increased growth in the presence of 16HBE$^{rs35699176}$ cells using the same infection model (Fig. 2f).

To investigate the cause of differences in *A. fumigatus* adhesion germination and growth in the presence of the variant cell line we next measured the secretion of soluble factors and spore uptake amongst 16HBE and 16HBE$^{rs35699176}$ cells. *A. fumigatus* germination rates in cell-free culture supernatants from 16HBE$^{rs35699176}$ and 16HBE cell lines were also measured. *A. fumigatus* spore germination was 7.83%, 12.87% and 19.65% higher in 16HBE$^{rs35699176}$ than in 16HBE at 4-, 5- and 6-h incubation, respectively ($P < 0.001$) (Fig. 2g). Linear regression analysis of the germination time points in cell-free-culture supernatants from 16HBE and 16HBE$^{rs35699176}$ cells replicated these observations (Supplementary figure 2). Interestingly, spores incubated with 16HBE$^{rs35699176}$ culture supernatants that had been heat-treated showed germination rates identical to spores

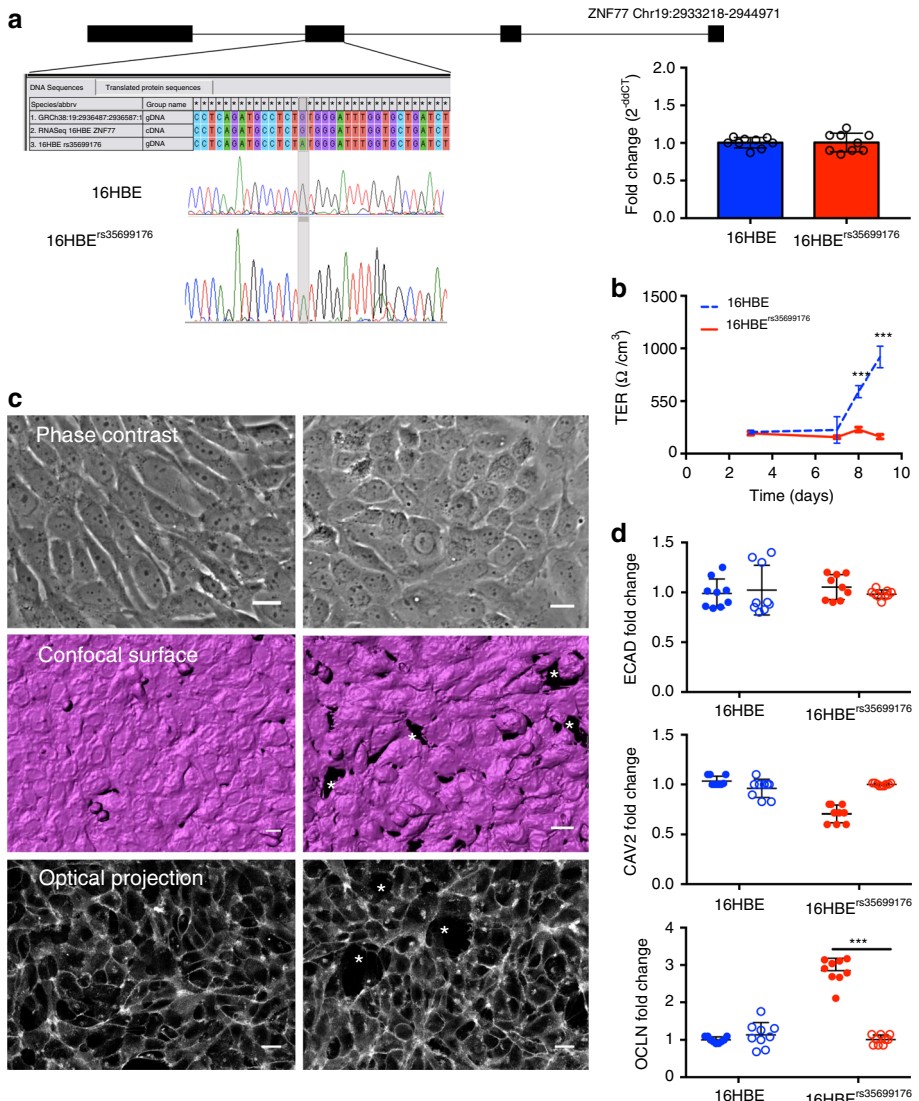

**Fig. 1** Altered morphology and confluence of bronchial epithelial cells carrying rs35699176. **a** Schematic of the rs35699176 targeted region and sequences of the human ZNF77 locus, including the chromatogram derived from Sanger sequencing of the mutant allele. Gene expression of ZNF77 in 16HBE cells and 16HBE$^{rs35699176}$ cell lines is equivalent (right panel). **b** Transepithelial electrical resistance (TER) does not increase after 8 days in monolayers formed by 16HBE$^{rs35699176}$ cells (panel **b**, solid line, \*\*\*$P < 0.0001$) indicating failure to form confluent cell layers. **c** Microscopy reveals alterations in 16HBE$^{rs35699176}$ monolayer morphology. Phase contrast microscopy shows rounded, non-adherent cells of 16HBE$^{rs35699176}$; confocal, surface rendered projection, stained with Cell Mask Deep Red shows holes in the monolayer and single optical section from projection. Images were processed using Metamorph and IMARIS v8.0.1 software. Asterisks denote lack of confluency. Scale bar represents 20 μm. **d** Fold change in gene expression of caveolin, occludin and E-cadherin genes after 6 h (solid columns) and 6 h (hashed columns) in 16HBE and 16HBE$^{rs35699176}$ cells co-cultured with *A. fumigatus*. Data in graphs are represented as mean ± standard deviation of a minimum of three experiments performed in biological and technical triplicates. Differences in gene expression and transepithelial resistance for each time point were analysed using the Mann–Whitney *U* test, \*\*\*$P < 0.001$

incubated with 16HBE culture supernatants (5%, 26% and 41% at 4-, 5- and 6-h co-incubation, respectively) suggesting the involvement of a protein factor mediating this phenotype (Supplementary figure 3).

Bronchial epithelial cells can act as non-professional phagocytes and internalize conidia via endocytosis and kill them via phagolysosome acidification. We found 16HBE$^{rs35699176}$ cells internalized *A. fumigatus* conidia more efficiently at 2 h post-infection than 16HBE cells (27.17% vs 37.5%, $P < 0.0001$) (Fig. 2h). This correlates with a higher percentage of adhesion to the genome-edited cell line. Germination of *A. fumigatus* spores results in exposure of spore surface proteins and carbohydrates that are recognized by

host cell receptors[25]. However, the percentage of spore uptake was similar for 16HBE and 16HBE$^{rs35699176}$ at later time points where germination had occurred in both systems suggesting that phagocytosis is ultimately similar in variant and parental cell lines.

We hypothesised that earlier *A. fumigatus* germination and growth could affect cytokine secretion and measured cytokine response at 6 h post-inoculation for 16HBE and 16HBE$^{rs35699176}$. *A. fumigatus* exposure induced higher levels of IL-6 and IL-10 secretion in 16HBE$^{rs35699176}$ cells than in 16HBE cells. The stronger cytokine induction in 16HBE$^{rs35699176}$ cells could be due to a response to the earlier formation of *A. fumigatus* germlings and hyphae previously observed (Fig. 2i).

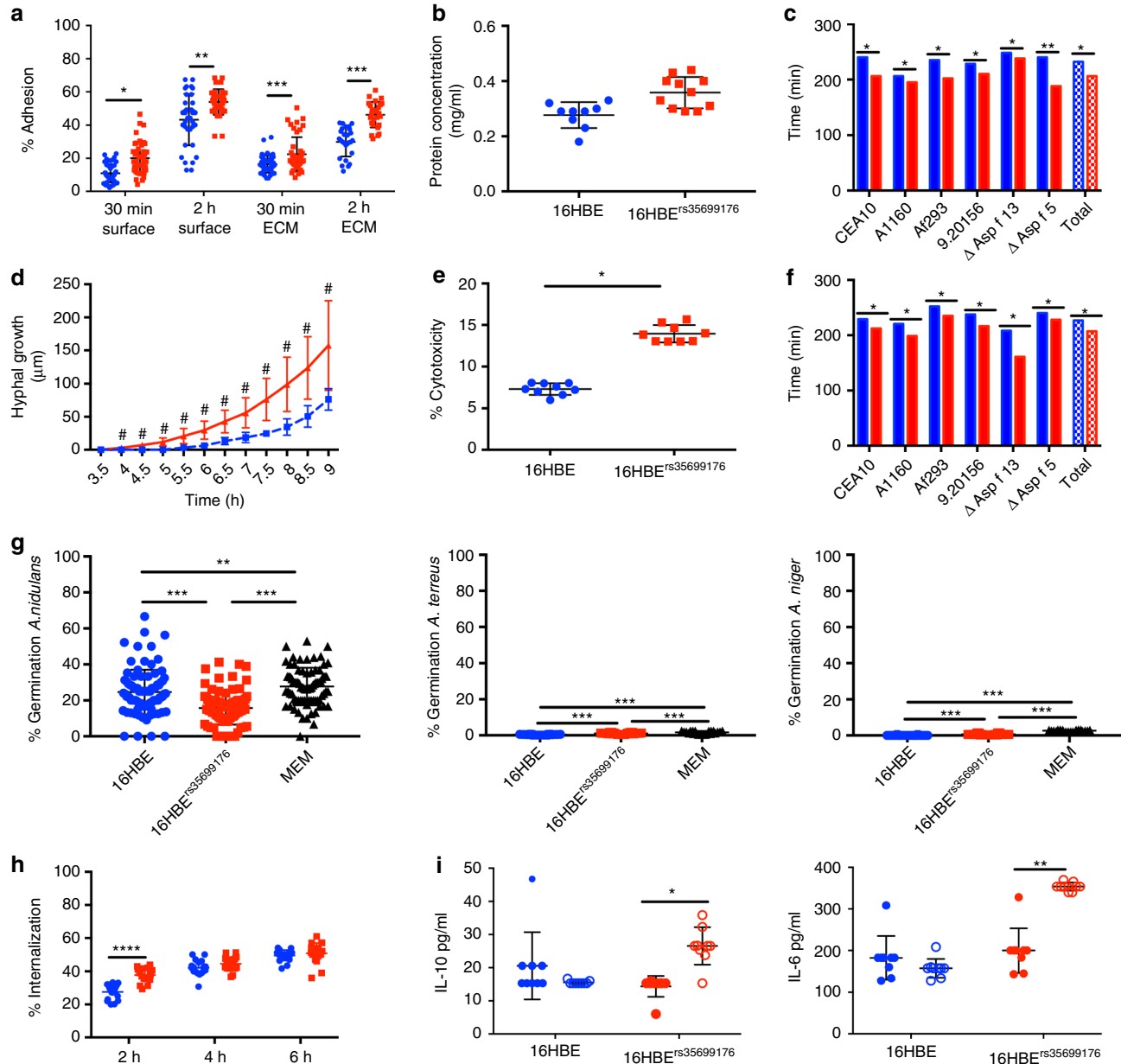

**Fig. 2** Changes in the integrity of 16HBE[rs35699176] epithelial monolayers influence *A. fumigatus* infection. Blue: 16HBE; Red: 16HBE[rs35699176]. **a** *A. fumigatus* conidia shows greater adhesion to cells and extracellular matrix components of 16HBE[rs35699176] cell lines at 30 min and 2 h after infection as determined by using two-way ANOVA with Sidak multiple comparison tests. **b** Extracellular matrix protein concentration is higher in 16HBE[rs35699176] than 16HBE cells. Differences between protein concentrations were performed using an unpaired *T*-test after performing normality test using D'Agostinho and Pearson normality test. **c** Germination time for *A. fumigatus* conidia inoculated onto 16HBE[rs35699176] (red) monolayers compared to 16HBE (blue) is decreased in the variant cell line. Differences were analysed using two-way ANOVA with multiple comparison test. **d** *A. fumigatus* CEA10 hyphal extension is increased in 16HBE[rs35699176] compared to 16HBE cells. Differences were analysed using an unpaired *T*-test after normal distribution being confirmed using the D'Agostinho and Pearson normality test. **e** Lactate dehydrogenase (LDH) assays show increased *A. fumigatus* CEA10 cytotoxicity in 16HBE[rs35699176] after 20 h infection. Differences were analysed using a non-parametric Mann–Witney test post-D'Agostinho and Pearson normality test. **f** Germination time of *A. fumigatus* spores in conditioned culture supernatant is reduced in 16HBE[rs35699176] cell lines (red) at 6 h infection time. Differences were analysed using a two-way ANOVA with multiple comparison test. **g** Significantly reduced spore germination of *A. nidulans*, *A. flavus* and *A. terreus* in 16HBE and 16HBE[rs35699176] cell supernatants compared to *A. fumigatus* (Supplementary Figure 2). Differences were analysed using one-way ANOVA with Dunnet's multiple comparison test. **h** *A. fumigatus* conidia are more effectively internalized by cells carrying the variant rs35699176 than in 16HBE controls at 2 h co-culture. Differences were analysed using two-way ANOVA with multiple comparison test. **i** Increased IL-10 and IL-6 release in 16HBE[rs356999176] cells in response to *A. fumigatus* exposure for 6 h. Transparent fill denotes *A. fumigatus* exposure. Differences were analysed using two-way ANOVA with multiple comparison test. Data in graphs are represented as mean ± standard deviation. Means were compared; *$P < 0.05$. **$P < 0.001$, ***$P < 0.0001$

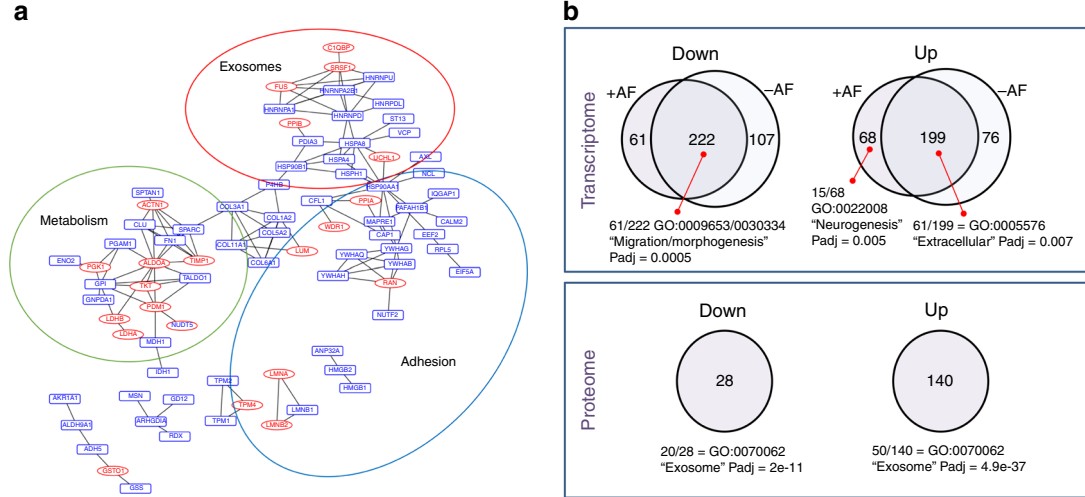

**Fig. 3** Secretome and RNA-seq analysis of 16HBE$^{rs35699176}$ show altered profiles in adhesion molecules. **a** Analysis of protein–protein interaction networks demonstrates higher secretion in the 16HBE$^{rs35699176}$ cell line of proteins involved in cell metabolism regulation and exosome pathways. Red ellipses represent proteins detected directly by mass spectrometry while the blue rectangles are proteins induced by the bioinformatics modelling within each network. **b** Functional category enrichment analysis of the differentially expressed and secreted proteins in 16HBE and 16HBE$^{rs35699176}$ bronchial epithelial with and without *A. fumigatus* challenge. Experiments were performed in biological replicates. Gene Ontology analyses were performed using strings. Statistical significance was calculated using strings

**ZNF77 regulates extracellular matrix expression**. To investigate the role played by proteins secreted by bronchial epithelial cells in controlling *Aspergillus* germination, we characterized the secretome of serum-starved cell culture supernatants from 16HBE and 16HBE$^{rs35699176}$ cells. Protein levels were measured using LC-MS/MS revealing a total of 234 significantly differentially secreted proteins (Supplementary figure 4, Supplementary data 1). After filtering ($\log_2$ fold change > 2 and $P < 0.05$), comparative protein abundance analysis reveals 140 proteins upregulated and 21 proteins down-regulated in 16HBE$^{rs35699176}$. Network and ontology analysis of the dysregulated proteins in 16HBE$^{rs35699176}$ demonstrated them to be involved in cell adhesion (GO:0044421, $P_{adj} = 1.63e{-}18$), metabolism (GO:00463464, GO:0044712, GO:0009132, $P < 0.0001$) and exosome (GO:0070062; $P_{adj} = 2.19e{-}22$) pathways. All of these are known to be important in regulating the immune response, cell remodelling and signalling in asthma[26–28]. Gene Ontology (GO) analyses demonstrated a significant enrichment of extracellular exosome terms (GO:0070062) for 20/28 down-regulated proteins and 50/140 up-regulated proteins in 16HBE$^{rs35699176}$ ($P_{adj} < 2e{-}11$) (Fig. 3a, b). Interestingly, proteins facilitating pathogen binding such as collagens, ficolins, cofilin and lectin binding proteins were more highly represented in 16HBE$^{rs35699176}$ and could credibly explain the increased *A. fumigatus* adhesion and spore uptake phenotypes of 16HBE$^{rs35699176}$ cells. However, an observed downregulation of proteins involved in oxidative processes such as lactotransferrin and calmodulin could be important in limiting pathogen clearance in the cell line carrying the rs35699176 genotype. We were not able to observe any significant differences in secretome composition between 16HBE and 16HBE$^{rs35699176}$ at 6 h post-inoculation with *A. fumigatus*.

Since the effect of *A. fumigatus* inoculation on epithelial cells at 6 h could be solely transcriptional we then performed RNAseq analysis on 16HBE and 16HBE$^{rs35699176}$ cells after 6 h *A. fumigatus* exposure (Supplementary data 2, 3, 4 and 5). Five of the most strongly regulated genes were validated using qRT-PCR (Supplementary figure 3). After interaction analysis in DESEQ2, we found that inoculation caused the same transcriptional changes in 16HBE and 16HBE$^{rs35699176}$ cells but that the transcriptomes of the two un-inoculated cell lines were significantly different.

GO analysis of the down-regulated genes in 16HBE$^{rs35699176}$ cell lines demonstrated a significant enrichment of genes involved in migration and morphogenesis (61/222 genes, GO:0009653 and GO:0030334, $P_{adj} = 0.0005$) that might be associated with the altered epithelium structure. Up-regulated genes in 16HBE$^{rs35699176}$ compared to 16HBE cells, were enriched for extracellular region components (GO:0005576, $P_{adj} = 0.007$). Comparison of difference in gene expression between 16HBE cells and 16HBE$^{rs35699176}$ cells at 6 h *A. fumigatus* infection, demonstrated an enrichment on genes involved in neurogenesis (15/68 genes GO:0022008, $P_{adj} = 0.005$).

We found a significant ($P_{adj} < 0.001$) enrichment of components leading to the activation of the immune response in 16HBE controls vs 16HBE inoculated with *A. fumigatus* including: Fc receptor mediated inhibitory signalling pathway (GO:0002767), positive regulation of interleukin-23 production (GO:0032747), positive regulation of NAD(P)H oxidase activity (GO:0033864), phagosome acidification (GO:0090383) or regulation of I-kappaB kinase/NF-kappaB signalling (GO:0043122). However, no changes in gene expression between uninfected 16HBE$^{rs35699176}$ and 6 h *A. fumigatus* challenged 16HBE$^{rs35699176}$ cells were observed suggesting a permissive role of the bronchial epithelium to *A. fumigatus* in patients carrying the rs35699176 genetic variant (Supplementary data 3 and 4).

**rs35699176 promotes airway colonization**. We explored the correlation between the rs35699176 genotype and fungal colonization in two different cohorts of patients with known fungal loads in respiratory samples (Tables 1 and 2). *A. fumigatus* burdens, measured by genome equivalents, were 15 times higher in bronchoalveolar lavage fluid from patients carrying rs35699176$^{+/-}$ ($P < 0.05$). Sixty percent (4/6 vs 0/15) of patients with >100 *A. fumigatus* genomes per ml of BAL carried the rs35699176$^{+/-}$ ($P = 0.0006$) (Fig. 4d).

**Table 1 Demographics of patients with known fungal loads determined by mycobiome analysis**

| | Total | rs35699176$^{-/-}$ | rs35699176$^{+/-}$ | P-value |
|---|---|---|---|---|
| ABPA (**n** = 7) | | | | |
| BMI | 25.71 ± 2.64 | 25.62 ± 2.73 | 25.78 ± 2.57 | ns |
| Smokers or ex-smokers | 14.28% | 0% | 25% | ns |
| Forced expiratory volume (FEV, l) | 1.64 ± 0.6 | 1.31 ± 0.1 | 1.80 ± 0.55 | ns |
| Forced vital capacity (FVC, l) | 3.11 ± 0.85 | 2.91 ± 0.62 | 3.21 ± 0.95 | ns |
| FEV/FEC (%) | 54.78 ± 14.08 | 50.31 ± 7.51 | 58.14 ± 16.65 | ns |
| Total IgE > 1000 IU | 28% | 33% | 25% | ns |
| log10 Aspergillus genome equivalents | 2.47 ± 1.04 | 1.90 ± 0.62 | 2.90 ± 0.94 | 0.0001 |
| SAFS (**n** = 6) | | | | |
| BMI | 25.7 ± 4 | 25.7 ± 4 | nf | nf |
| Smokers or ex-smokers | 16.66% | 16.66% | nf | nf |
| Forced expiratory volume (FEV, l) | 2.14 ± 0.81 | 2.14 ± 0.81 | nf | nf |
| Forced vital capacity (FVC, l) | 3.71 ± 1.6 | 3.71 ± 1.6 | nf | nf |
| FEV/FEC (%) | 56.13 ± 9.98 | 56.13 ± 9.98 | nf | nf |
| Total IgE > 1000 IU | 0% | 0% | nf | nf |
| log10 Aspergillus genome equivalents | 1.55 ± 0.55 | 1.55 ± 0.55 | nf | nf |
| Asthmatics (**n** = 4) | | | | |
| BMI | 28 ± 2 | 28 ± 2 | nf | nf |
| Smokers or ex-smokers | 25% | 25% | nf | nf |
| Forced expiratory volume (FEV, l) | 2.20 ± 1.02 | 2.20 ± 1.02 | nf | nf |
| Forced vital capacity (FVC, l) | 3.52 ± 1.02 | 3.52 ± 1.02 | nf | nf |
| FEV/FEC (%) | 59.65 ± 9.75 | 59.65 ± 9.75 | nf | nf |
| Total IgE > 1000 IU | 0% | 0% | nf | nf |
| log10 Aspergillus genome equivalents | 1.27 ± 0.31 | 1.27 ± 0.31 | nf | nf |
| Healthy controls (**n** = 4) | | | | |
| BMI | 32 ± 9 | 32 ± 9 | nf | nf |
| Smokers or ex-smokers | 0% | 0% | nf | nf |
| Forced expiratory volume (FEV, l) | n/a | n/a | nf | nf |
| Forced vital capacity (FVC, l) | n/a | n/a | nf | nf |
| FEV/FEC (%) | n/a | n/a | nf | nf |
| Total IgE > 1000 IU | 0% | 0% | nf | nf |
| log10 Aspergillus genome equivalents | 1.05 ± 0.28 | 1.05 ± 0.28 | nf | nf |

Mean and standard deviation is shown
*FVC* forced vital capacity, *FEV1* forced expiratory volume measured during the first forced breath, *nf* not found

**Table 2 Demographics of patients with ABPA and known fungal loads determined by *Aspergillus*-specific qPCR**

| | Total | rs35699176$^{-/-}$ (n = 36) | rs35699176$^{+/-}$ (n = 9) | P-value |
|---|---|---|---|---|
| BMI (kg/cm$^2$) | 26.59 ± 3.69 | 26.58 ± 3.52 | 26.75 ± 2.62 | ns |
| Smokers or ex-smokers (%) | 35% | 36% | 33% | ns |
| BTS score (median) | 4 | 4 | 4 | ns |
| Forced expiratory volume first second (FEV1, l) | 1.83 ± 0.79 | 1.83 ± 0.80 | 2.07 ± 0.75 | ns |
| Forced vital capacity (FVC, l) | 3.18 ± 1.02 | 3.2 ± 1.03 | 3.2 ± 0.98 | ns |
| FEV1/FVC (%) | 57 ± 13 | 56 ± 12 | 58 ± 14 | ns |
| Total IgE > 1000 IU (%) | 37.7% | 36.10% | 44.40% | ns |
| *Aspergillus* PCR (Ct) | 37.37 ± 5.08 | 37.57 ± 2.7 | 35.7 ± 1.1 | 0.01 |
| % Positive PCR | 71.1% | 63% | 100% | 0.04 |

Mean and standard deviation is shown
*BTS* British thoracic score, *FVC* forced vital capacity, *FEV1* forced expiratory volume measured during the first forced breath

As it is not possible to systematically bronchoscope this patient group we then investigated the association of rs35699176 with fungal burden in sputum measured using qPCR. Forty-five patients with ABPA were tested by qPCR (Fig. 4e). Twenty-eight percent of patients (9/32) with an *Aspergillus* positive sputum PCR and 0% (0/13) patients with an *Aspergillus* negative sputum PCR carried rs35699176$^{+/-}$ ($P = 0.036$). *Aspergillus*-specific qPCR Ct values in rs35699176$^{+/-}$ patients were significantly higher than in those with rs35699176$^{+/+}$, indicating that rs35699176 doubles fungal loads in sputum derived from the upper respiratory airways. Asthma severity measured by BTS

score, FEV1 and FVC were not significantly different between groups (Table 1).

## Discussion

Increased *A. fumigatus* colonization of the airway epithelium is associated with worse outcome in patients with fungal disease[29]. Given that exposure to *A. fumigatus* spores is constant it has been theorised that colonization and infection are determined by host genetic factors. In the case of ABPA, this phenotype could be partially explained by the impaired airway epithelium function in

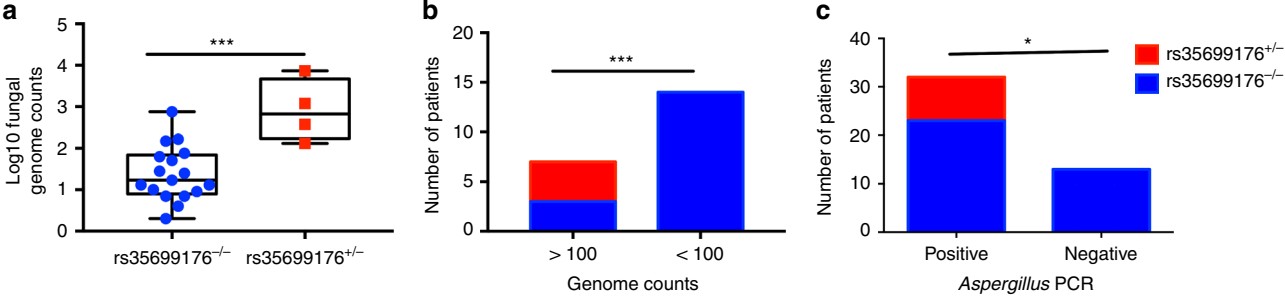

**Fig. 4** Individuals heterozygous for rs35699176 have higher lung *A. fumigatus* burden. **a**, **b** Correlation between rs35699176[+/−] and fungal load in patient bronchoalveolar samples with fungal loads expressed as genome counts. Boxplots represent median and maximum and minimum values. Comparisons were performed using Fisher exact test. **c** Association between a positive *Aspergillus* PCR in sputum and rs35699176[+/−]. Comparisons were performed using Fisher exact test. *$P < 0.05$, **$P < 0.001$, ***$P < 0.0001$

asthma but not all asthmatic patients develop ABPA[10]. We observed that a genetic variant, rs35699176, causing a premature stop codon in ZNF77 leads to a pathophysiological mechanism whereby *A. fumigatus* colonization of the respiratory airways is increased. Bronchial epithelial cells carrying rs35699176 showed strongly altered levels of cell matrix components and were more permissive to *A. fumigatus* promoting greater adhesion, germination and growth. Importantly this observation was validated in patients with ABPA where strongly increased fungal burdens were associated with the presence of the variant, suggesting the utility of rs35699176 genotyping as a colonization risk-marker in ABPA.

Compromised epithelial barrier function is associated with the development of atopic asthma possibly due to increased tissue penetrance of allergens and inflammatory factors[30,31]. The lack of confluence of cells carrying the rs35699176 genetic variant could facilitate passage of *A. fumigatus* antigens into the airway tissue, leading to immune activation and ultimately to expression of ABPA. Tight junction and adhesion proteins provide structural integrity to the epithelial barrier, block allergen influx, provide resistance to pathogens and control cell homeostasis[22]. A wide number of allergic diseases are exacerbated by a loss of epithelial barrier function associated to changes in the expression of filaggrin[32], claudin-1[33], adhesion and tight junction proteins[34,35].

Interestingly, occludin was upregulated in 16HBE[rs35699176] compared to 16HBE cells but it was rapidly downregulated after *Aspergillus* challenge. This suggests that 16HBE[rs35699176] cells might over-express occludin to try to compensate for the downregulation of other genes involved in junction formation (e.g. GJB3, CDH1, PCDHGA4) and morphogenesis (e.g. TMEM217, TGM2, IGF2). *A. fumigatus* spores undergo several morphological changes during their interaction with the lung epithelium[36]. Early after inhalation, *A. fumigatus* conidia will contact the airway epithelium where they swell and germinate to form hyphae. Native and swollen spores can be internalized before undergoing germination and hyphal growth within epithelial cells[23]. Moreover, factors secreted by epithelial cells (defensins, PTX3, etc.) control the initial steps of the *A. fumigatus* clearance response[37,38].

Our results indicate that an upregulation of gene expression and the synthesis of proteins involved in the extracellular region of the cells such as collagens, ficolin, lectins, laminin or integrins in 16HBE[rs35699176] cells facilitates an earlier *A. fumigatus* adhesion to the host surface[26,39–41]. Those components can act like a glue to anchor the fungal spore to host. Lung epithelial cells are recognized as non-professional phagocytes and can contribute to spore clearance via endocytosis[27,28,42]. Earlier adhesion to the host surface in rs35699176[+/+] cells correlates with an enhanced spore uptake from those cells 2 h after exposure. However spore

phagocytosis is the same for parental and variant cell lines after 4 h. Spore internalisation is dependent on two crucial regulators of the actin dynamics of the host, phospholipase D[43] and cofilin-1[42], which are activated after exposure to 1,3-β-D glucan on the surface of swollen or germinating conidia and hyphae. We found that cofilin-1 and ARC were significantly overexpressed in unchallenged 16HBE[rs35699176+/+] cells and this might explain why 16HBE[rs35699176+/+] cells could more effectively phagocytose fungal spores at 2 h but not at later time points[44]. In this model, the increased fungal burden arises from early germination of non-internalized spores rather than from escape from phagocytosis. No host-derived growth promotion factors have been described previously for moulds although factors modulating bacterial and *Candida* growth have been observed[45–47]. The only host factors described that might modulate *Aspergillus* germination and growth are limited to human glycogen synthase kinase 3 activity, corticosteroid treatment or smoking status[13,48,49].

The increase in adhesion and germination caused by rs35699176 is specific to *A. fumigatus* and is not observed for other closely related pathogenic *Aspergilli*. *A. fumigatus* is the main cause of fungal allergy but the other *Aspergillus* species are also frequently found in the environment and cause the same spectrum of allergic, chronic and invasive diseases[2]. The lack of germination and growth responses in the other *Aspergillus* species might explain the predominance of *A. fumigatus* in aspergillosis.

Bronchial epithelial cells carrying rs35699176 responded with a stronger inflammatory IL-6 and IL-10 response to *A. fumigatus* conidia after infection. Interestingly, those cytokines are released in response to fungal proteases released by growing hyphae[50,51]. The faster exposure of glucans and earlier hyphal development observed after inoculation of 16HBE[rs35699176] cells with *A. fumigatus* spores could explain why those cytokines can be detected in cell culture supernatants from 16HBE[rs35699176] cells but not the parental cell line. The faster production of cytokines may have an important role in determining fungal burden at later time points. It has recently described that a genetically conditioned overexpression of IL-10 predisposes to invasive aspergillosis by suppressing the anti-*Aspergillus* response of macrophages[52]. Additionally, high serum levels of IL-10 in patients with cystic fibrosis are associated with *Aspergillus* colonization and ABPA[53].

Although low-level colonization of the respiratory airways by *A. fumigatus* is universal[54], patients with ABPA have higher fungal loads in their respiratory airways. Antifungal treatment is not considered primary therapy in ABPA, but azole treatment is useful in decreasing fungal burdens and minimizing exacerbations[10,55] and a recent trial of azole therapy for severe asthma in the context of fungal sensitisation showed a strong effect on patient quality of life score. Here, we showed that patients

heterozygous for rs35699176 have higher fungal burdens in the upper respiratory airways in both BAL and sputum samples. Although this phenotype did not correlate with worse asthma outcome in our exploratory cohorts, patients with the rs35699176 variant frequently carry elevated levels of allergenic fungus and therefore could benefit from antifungal treatment.

Our results show that in normal individuals ZNF77 prevents fungal persistence in the airways by regulating epithelial integrity and controlling the secretion of proteins, which modulate pathogen adhesion to the bronchial epithelium and extracellular matrix[56,57]. The permissive response of the bronchial epithelium to *A. fumigatus* resulting from rs35699176 can contribute to chronic inflammation and the development of allergy through increased fungal burden. Although we show direct effects of rs35699176 genetic variant on fungal adhesion and growth on 16HBE cells, the effects of an altered epithelium in lung tissue are likely to be more complex as fungi will be able to access the basal lamina and other cell types will be present at sites of colonization or infection. The fact that we did not find patients homozygous for rs35699176 suggests that this could be lethal although the low frequency of the allele means that our study population is likely to be too small to observe homozygous individuals. Genome-editing tools available at the time this work was carried out did not allow for the recreation of heterozygous mutations. Moreover, a limitation of the epithelial cell model is that it does not currently allow longer-term experiments that might more accurately affect colonization. Hence the variant phenotype may only be accessible using homozygous variants. Longitudinal studies to correlate the rs35669176 genotype and *Aspergillus* disease are required to assess other factors that may affect colonization. Development of new strategies to modulate *A. fumigatus* adherence and colonization in ABPA might be a useful therapeutic avenue for individuals carrying the rs35699176 genotype.

## Methods

**Generation of 16HBE genome-edited bronchial epithelial cells**. To study the biological function of the genetic variant rs35699176 in fungal colonization discovered in a whole exome sequencing study[58] (see Supplementary note 1 and Supplementary table 1), we developed an in vitro model using genome-edited 16HBE bronchial epithelial cells[59]. 16HBE cells were a gift from Dr. D.C. Gruenert, University of California[60]. Mycoplasma contamination testing was performed by the University of Manchester cell culture facilities. To ensure that ZNF77 function and sequence alteration was correct in this cell line, we sequenced the region surrounding the variant and also performed variant calling from RNAseq data obtained in our laboratory as part of a previous study. Cell guides for targeting this region were designed by analysing a 1 kb genomic region surrounding the rs35699176 genetic variant (http://www.ensembl.org/index.html) and choosing Protospacer adjacent motif (PAM) sequences close to the genetic variant with ChopChop Software (https://chopchop.rc.fas.harvard.edu/). The selected CRISPR/Cas9 guide was directed to the genomic region Chr19: 2936525–2936547 as this region had no predicted off-target effects. Oligonucleotides containing the CRISPR/Cas9 guides were synthetized by Eurofins Genomics (Hamburg, Germany) (Supplementary Table 2). Guides were cloned into the GeneArt CRISPR Nuclease Vector with OFP Reporter Kit (Invitrogen, Paisley, UK) according to the manufacturer's instructions. Ten clones were sequenced by using U6 primers provided by the manufacturer. Control oligo included in the kit and pUC19 were used as transformation controls. Qiagen Midiprep plasmid kits (Qiagen, Manchester, UK) were used to isolate plasmid DNA.

To be able to introduce the specific genetic variant (rs35699176) in 16HBE cells, a single strand donor repair template (138 bp) was designed and purchased from IDT Technologies (Supplementary Table 2). The donor template contains the specific genetic variant and a non-sense mutation in the PAM sequence to avoid Cas9 cutting the repair template, for this purpose the TGG-PAM sequence was changed to TGA which is synonymous in ZNF77. Ultramer oligos were used at a 10 μM final concentration (IDT Technologies, Coralville, USA) (Supplementary Table 2).

16HBE cells were maintained in supplemented Minimum Essential Medium (MEM, Sigma, Poole, UK) with 10% FBS, 1% L-glutamine and 1% penicillin streptomycin. Cells were seeded at $2 \times 10^5$ cells per well in a 24-well plate and incubated at 37 °C, 5% $CO_2$ for 16 h. Before transfection, medium was removed completely and replaced with MEM containing 1% L-glutamine and cells were incubated for an extra 2 h at 37 °C, 5% $CO_2$. Each well was transfected with 500 ng of plasmid and 1 pmol repair template using Lipofectamine 3000 reagent according

to the manufacturer's instructions. Four hours after transfection, medium was changed with supplemented MEM. Toxicity controls were included in optimization experiments. Following transfection (48 h) cells were sorted and incubated for 2–3 weeks for clonal expansion. Those clones which were able to grow were scaled up in culture and analysed by sequencing. The insertion of the mutation was confirmed by sequencing a 663 bp region in ZNF77. Genome DNA from transfected cells was isolated by using Gentra Pure DNA kit (Qiagen, Manchester, UK) according to the manufacturer's instructions. PCR reactions were performed in 25 μl final volume PCR mix containing 1× MasterMix (Promega, Southampton, UK), 0.5 μM of each primer (rs35699176-3 and rs35699176-4) and 10 ng of DNA. Samples were amplified using the following cycling parameters: an initial cycle of 2 min at 95 °C, followed by 30 cycles of 30 s at 95 °C, 30 s at 58 °C, and 30 s at °68 C, with a final cycle of 10 min at 68 °C. Reaction products were analysed in a 2% agarose gel and purified using QIAquick PCR Purification Kit (Qiagen, Manchester, UK) and sequenced by Eurofins genomics. The ZNF77 gene sequence available in Ensembl GrCh38 and the sequence of the ZNF77 gene in a 16HBE background were used as reference. The effect of the insertion of the rs35699176 genetic variants on ZNF77 gene expression was also evaluated. RNA was extracted using the RNAeasy kit (Qiagen, Manchester, UK). ZNF77 gene expression was evaluated by using primers ZNF77_2934825 and ZNF_2936494 which allowed for specific amplification of 100 bp of ZNF77 mRNA (Supplementary Table 2). The GAPDH gene was used as a housekeeping gene (Supplementary Table 2). Reactions were performed in a 20 μl final volume PCR mix containing 1× SensiFASTM SYBR® Hi-ROX One-Step Mix (Bioline Reagents, London, UK), 0.2 μl of reverse transcriptase, 0.4 μl of ribosafe inhibitor, 500 nM of each primer and 40 ng of RNA. For reverse transcription reactions, samples were incubated for 10 min at 50 °C. Then, for PCR amplification, samples were denatured for 2 min at 95 °C followed by 40 cycles of 5 s at 95 °C, 30 s at 55 °C for ZNF77 amplification or 60 °C for GAPDH amplification, and finally 72 °C for 30 s. A final gradient step from 65 to 95 °C (5 s per step) was included. Gene expression was analysed using the ddCt method and differences in gene expression were compared using Mann–Whitney *U* test (GraphPad 7.0, La Jolla, CA). Experiments were performed in biological and technical triplicates. Data is represented in the manuscript as mean and standard deviation unless other stated.

**Evaluation of epithelial barrier function**. 16HBE and 16HBE[rs35699176] cells were seeded into 1 μm Transwell inserts (Corning, USA) at a density of $10^6$ cells in 12-well plates. Transepithelial resistance (TER) was evaluated using a EVOM2™ Epithelial Volt/Ohm Meter (World Precision Instruments, USA). Experiments were performed at least in biological and technical triplicates. Differences between 16HBE and 16HBE[rs35699176] cells were compared for each time point using a Mann–Whitney *U* test (GraphPad, La Jolla, CA).

Results were confirmed by confocal and phase contrast microscopy. For that, 16HBE and 16HBE[rs35699176] cells were seeded at $1 \times 10^5$ cells in a 2-well Idibi chamber (Thistle Scientific, Glasgow, UK) and incubated for 48 h at 37 °C, 5% $CO_2$. Z-stacks of confocal images were acquired using a fully motorized Leica SP8 laser scanning confocal microscope equipped with a 63XHC PlanApo UVIS CS2 water immersion (NA 1.2) objective. Imaging was performed at 37 °C in a temperature controlled chamber. Monolayers stained with the plasma membrane Cell Mask Red (ThermoFisher, UK) were excited with a tunable white light laser (70% power) at 649 nm and fluorescence was detected between 660 and 750 nm. Z-stacks of confocal images were surface rendered using IMARIS v8.0.1 Image Processing and Visualization software (Bitplane, Switzerland). For phase contrast microscopy, cells were imaged using a Nikon TE-2000E microscope and processed using Metamorph software.

**Aspergillus strains**. *Aspergillus* strains (Supplementary Table 3) were cultured at 37 °C for 48–72 h on Sabouraud dextrose agar (Oxoid, Basingstoke, UK). Spores were harvested using PBS-containing Tween 20 at 0.1%. Spore stock solutions were washed twice on supplemented MEM without FBS.

**A. fumigatus-bronchial epithelial cells co-culture**. 16HBE and 16HBE[rs35699176] bronchial epithelial cells were maintained at 37 °C, 5% $CO_2$ in supplemented MEM. Epithelial cells were used after the second or third passage for each experiment. For all experiments, $2 \times 10^5$ 16HBE and 16HBE[rs35699176] cells were seeded in 24-well plates and incubated for 16 h (confluence > 90%). Cells were then washed twice with supplemented MEM without FBS and cells were serum-starved for 48 h. Monolayers were challenged with $2 \times 10^4$ *A. fumigatus* spores (0.1 MOI) and incubated for 6 h (Supplementary table 3). Zero infection controls were made by mock inoculation using supplemented MEM without FBS. After incubation, cell culture supernatants were stored at −80 °C until needed and RNA was extracted using the RNAeasy kit (Qiagen).

**qPCR**. Validated primers (Supplementary Table 2) to measure the expression of E-cadherin (ECAD), occludin (OCLN) and caveolin (CAV2) were purchased from Biorad (Biorad, UK). qPCR amplifications were performed according to manufacturer's instructions and using SensiFASTM SYBR according to manufacturer's instructions. Gene expression was analysed using the ddCt method and differences

in gene expression were compared using Mann–Whitney $U$ test (GraphPad 7.0, La Jolla, CA). Experiments were performed in biological and technical triplicates.

**LDH assay.** The Pierce LDH cytotoxicity assay kit (ThermoFisher Scientific, Rockford, USA) was used to determine *A. fumigatus*-mediated cytotoxicity assay according to the manufacturer's instructions. Samples were assessed in technical and biological triplicates. Briefly, $10^4$ 16HBE and 16HBE[rs35699176] bronchial epithelial cells were seeded in a 96-well plate and incubated for 16 h until confluence. Cells were serum-starved for 48 h as described above and infected with $10^4$ *A. fumigatus* CEA10 spores. Differences in cell toxicity mediated by *A. fumigatus* were determined at 6, 16 and 20 h post-infection and analysed by $T$-test after normal distribution was confirmed (GraphPad Prism 7.0).

**Time-lapse microscopy.** Cells were seeded at $2 \times 10^5$ cells per well in a 24-well glass bottom plates (Corning, UK) and incubated for 16 h until confluent. Cells were serum-starved for 48 h and then infected with $2 \times 10^4$ spores from the *A. fumigatus* strains described in Supplementary Table 3. The plate was then mounted under a Leica SP8X inverted confocal microscope using a $40 \times$ dry lens objective in a 5% $CO_2$ environment at 37 °C. Three different positions per well in 3 independent biological and technical replicates were evaluated. Images were captured using the brightfield transmitted light detector of the microscope every 30 min for 11 h driven by the Leica LASAF image software. The videos generated by this software were exported as AVI documents and processed with Image J software (NIH) (http://rsb.info.nih.gov/ij). Hyphal length, *A. fumigatus* germination time point and total growth were measured using Fiji Software (https://fiji.sc/). Differences in *A. fumigatus* hyphal length and germination time point were determined by $T$-test at fixed time points after confirming normal distribution (GraphPad Prism v7). Differences in *A. fumigatus* growth curves were calculated using the non-parametric Kruskal–Wallis test using GraphPad Prism 7 software. Experiments were performed in biological and technical triplicates.

**Binding of *A. fumigatus* spores.** 16HBE and 16HBE[rs35699176] cells were seeded at $2 \times 10^5$ cells per well in a 24-well plate for 24 h. Plates were washed once with PBS. Cells were incubated for 1 h at 37 °C in MEM + 1% L-Glutamine + 1% BSA and incubated for 1 h at 37 °C to block receptors that could overshadow cell adhesion[39]. After that, medium was replaced with 1 ml MEM + 1% L-glutamine containing $2 \times 10^5$ *A. fumigatus* CEA10 spores. The percentage adhesion was determined after 30 min and 2 h incubation time. Cells were then fixed in 3% paraformaldehyde in PBS for 10 min. Spores were counted using a Nikon TE-2000E microscope with a dry 20× objective. Five pictures per well in biological and technical triplicates were processed using FIJI software and results were expressed as percent adhered spores. Differences in the percentage of spores binding between 16HBE and 16HBE[rs35699176] were determined for each time point using two-way ANOVA with multiple comparison test using GraphPad Prism 7.

**Adhesion of *A. fumigatus* spores to the extracellular matrix.** Cells were seeded at $2 \times 10^5$ cells per well in a 24-well plate for 24 h and then treated as described in ref. [39]. Quantification of the extracellular matrix layer was performed using a BCA assay (Sigma-Aldrich, Poole, UK). The percentage of adhesion after 30 min and 2 h incubation was performed as described before. Differences in protein concentration between cell lines was determined by using an unpaired $T$-test after performing normality test using D'Agostinho and Pearson normality test in GraphPad Prism 7.

**Cytokine release.** Cytokine quantification produced by 16HBE and 16HBE[rs35699176] bronchial epithelial cells in the presence of *A. fumigatus* CEA10 was determined by using the LEGENDplex™ Human Inflammation Panel (BioLegend, London, UK) and a BD Influx cell sorter machine. This panel allows simultaneous quantification of 13 human inflammatory cytokines/chemokines, including IL-1β, IFN-α, IFN-γ, TNF-α, MCP-1 (CCL2), IL-6, IL-8 (CXCL8), IL-10, IL-12p70, IL-17A, IL-18, IL-23 and IL-33. Data was analysed by using LEGENDplex data analysis software (http://www.biolegend.com/legendplex/software). Differences in cytokine concentration at 6 h *Aspergillus* challenge were performed by using the two-way ANOVA with multiple comparison test (GraphPad, Prism 7.0, La Jolla, CA) in experiments run in biological and technical triplicates. Statistically significant differences were accepted for $P < 0.05$.

**Germination assay in a cell-free environment.** Cells were seeded at $2 \times 10^5$ cells per well in a 24-well plate and incubated for 16 h at 37 °C, 5% $CO_2$. Cells were then serum-starved for 48 h and supernatants were collected, centrifuged for 3 min at 1200 rpm to remove cell debris and stored at −80 ºC until use. For analysis, samples were thawed on ice and transferred to a 24-well glass bottomed plate and infected with $2 \times 10^4$ spores of the *Aspergillus* strains. Percentage germination was estimated at 4, 5 and 6 h post-inoculation at 37 C, 5% $CO_2$ using a total of 15 images per well by using a Nikon TE-2000E microscope with a 63× lens and processed using Metamorph software. Percentage spore germination was estimated using the cell counter application in Fiji software. Experiments were performed in technical and biological triplicates including germination in MEM serum-free

media as control. Differences in the percentage of germination were calculated by one-way ANOVA with multiple comparison test using GraphPad Prism 7 software.

**Internalization assay.** 16HBE and 16HBErs35699176 cells were seeded in 24-well glass bottom plates as described. Cells were then challenged with $10^5$ *A. fumigatus* CEA10 spores-GFP in serum-free media and, incubated for 2, 4 and 6 h at 37 C, 5% $CO_2$. Infected monolayers were stained with Cell Mask as previously described. Cells were then fixed with 3% formaldehyde and 3D images were obtained using a confocal laser scanning microscope. Spores inside and outside the epithelial monolayer were automatically counted using IMARIS v8.0.1 (Bitplane) software. Statistical analyses were performed by two-way ANOVA with multiple comparisons test using GraphPad Prism. Experiments were performed in biological and technical triplicates.

**Heat-inhibition assays.** Cell culture supernatants were heat-treated for 10 min at 95 °C and then cooled at room temperature. Germination experiments were performed using *A. fumigatus* CEA10 as described above. Statistical analyses were performed by two-way ANOVA with multiple comparisons using GraphPad Prism. Experiments were performed in biological and technical triplicates.

**Secretome analysis.** 16HBE and 16HBE[rs35699176] cells ($1 \times 10^6$ cells per well) were seeded in 6-well plates for 24 h at 37 °C and 5% $CO_2$. Cells were then washed 10 times with 3 ml serum-free medium and incubated for 48 h. At the time of the experiment, medium was replaced with fresh 3 ml non-supplemented MEM and incubated for 6 h. Cell culture supernatants were then collected and centrifuged at 12,000 rpm for 30 min to remove cell debris. Experiments consisted of 3 biological replicates. Cell culture supernatants were prepared for mass spectrometry using a modified filter-aided sample preparation (FASP) method with modifications. Approximately 3 ml cell culture supernatants were concentrated to 20 μl using Microcon-30 kDa centrifugal filter units (Merck Millipore). Protein concentration was determined using a Millipore Direct Detect® spectrometer at AM1. LysC was used for the initial digestion at a ratio of 1:30 (enzyme:protein). After 3 h, the urea concentration was dropped from a concentration of 6 M to 1 M with the addition of Tris–HCl, pH 8. The proteins were then digested with trypsin at a ratio of 1:50 (enzyme:protein). After digestion, peptides were collected and the samples were desalted with OLIGO™ R3 reversed-phase media.

Digested samples were analysed by LC-MS/MS using an UltiMate® 3000 Rapid Separation LC (RSLC, Dionex Corporation, Sunnyvale, CA) coupled to an Q Exactive™ HF Hybrid Quadrupole-Orbitrap™ mass spectrometer (ThermoFisher Scientific, Waltham, MA, USA). Peptide mixtures were separated using a 90 min gradient, from 92% A (0.1% FA in water) and 8% B (0.1% FA in acetonitrile) to 33% B, in 66 min at 300 nl permin, using a 250 mm × 75 μm i.d. 1.7 mM BEH C18 analytical column (Waters, Wilmslow, UK). Peptides were selected for fragmentation automatically by data dependent analysis.

The acquired MS data was analysed using Progenesis LC-MS (v4.1, Nonlinear Dynamics, Newcastle upon Tyne, UK). Features with charges ≥+5 were masked and excluded from further analyses, as were features with less than 3 isotope peaks. The resulting peak lists were searched against the Uniprot Human database (version 20151111) using Mascot v2.5.1, (Matrix Science, Boston, USA). The Mascot results were imported into Progenesis LC-MS for annotation of peptide peaks. Network analysis and GO of dysregulated proteins was performed using String v9.0 (http://string-db.org). The induced network was further visualised by using Cytoscape v3.2.1.

**RNA-seq analysis.** Changes in the expression between 16HBE and 16HBE[rs35699176] cells exposed to *A. fumigatus* for 6 h at a 0.1 MOI were analysed by RNA-seq. RNA was extracted using RNAeasy kit (Qiagen, Manchester, UK). Experiments consisted of 3 biological replicates. Strand-specific RNA-seq libraries were prepared using the Illumina workflow with the TruSeq® Stranded mRNA Sample Preparation Kit (Illumina, San Diego, USA). Paired-end reads were generated from each sample. Up to 78 M of total reads were obtained from each sample. The FASTQ files generated by HiSeq Illumina 4000 platform were analysed with FastQC and any low quality reads and contaminated barcodes and primers were filtered and trimmed with Trimmomatic v3.2. All libraries were aligned to the hg19 ch37 assembly of human genome using TopHat-2.1.0 (https://ccb.jhu.edu/software/tophat/index.shtml) and only the best-mapped reads were collected for differential analysis. The mapped reads were counted with HTSeq against Gencode_v16.gtf (https://www.gencodegenes.org). Differentially expressed (DE) genes were identified between each pair of different levels cell type (16HBE and 16HBE[rs35699176]) with the DESeq2 package (https://bioconductor.org). GO enrichment analysis was performed using String v9.0 (http://string-db.org).

**rs35699176 genotyping.** DNA from 21 bronchoalveolar lavages from patients with known fungal loads (previously determined using metagenomics sequencing) (Local Research Ethics Committee REC reference no 11/NW/0175) was used for ZNF77 genotyping[13]. Patient demographics are shown in Table 1. Samples were sequenced as detailed in Generation of 16HBE[rs35699176] bronchial epithelial cells section using primers rs35699176-3 and rs35699176-4 and analysed using Chromas-Lite (http://chromas-lite.software.informer.com). Associations between

genotype and fungal load were analysed by *T*-test and FET using GraphPad Prism 7.0 (La Jolla, CA, USA).

For validation, DNA samples from sputum of 45 ABPA patients were used for genotyping (Table 2, ManArts M2016-45). *Aspergillus* burden in sputum samples was determined by *Aspergillus* real-time PCR (Progenie *Aspergillus* FUMI, Werfen, UK) at the National Aspergillosis Center (Manchester, UK). Correlation between genotype and PCR result was determined by Chi-square and Fisher exacts test using GraphPad Prism 7.0.

All patients in the study provided their informed consent according to the Biobank regulations.

## Data availability

All relevant data supporting the findings of the study are available in this article and its Supplementary Information files, or from the corresponding authors upon request, with restriction of data that would compromise patient confidentiality. The RNA-seq data used in this study have been deposited in the EMBL Arrayexpress platform under the accession number E-MTAB-7066 [https://www.ebi.ac.uk/arrayexpress/experiments/E-MTAB-7066]. Secretome data can be accessed in [https://www.figshare.com/s/adf8b6dbda50aab943fe]. https://doi.org/10.6084/m9.figshare.6833810.

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

## Acknowledgements

The research leading to these results has been funded by the Fungal Infection Trust. We would like to thank Dr. Ping Wang of the Bioinformatics Core Facilities at the University of Manchester for providing support analysing the RNA-seq data, Dr. Gareth Howell of the Flow Cytometry Core Facility at the University of Manchester for his assistance in conducting FACS analysis, Dr. Ronan O'Cualain, David Knight and Julian Selley of the Biological Mass Spectrometry Core Facility for their help in performing secretome data analysis, Lea Gregson for her support in cell culture and Darren Thomson for his help in assessing time-lapse microscopy experiments. We would also like to thank the Mycology Reference Centre Manchester for providing some of the strains used in this study and for performing *Aspergillus* PCRs, Charles Streuli and Martin Humphries for their critical comments, and Marcin Frazer and Livingstone Chishimba for providing the bronchoalveolar DNA samples. This report is independent research supported by the National Institute for Health Research Clinical Research Facility at University Hospital of South Manchester NHS Foundation Trust. The views expressed in this publication are those of the authors and not necessarily those of the NHS, the National Institute for Health Research or the Department of Health. The authors would like to acknowledge the Manchester Allergy, Respiratory and Thoracic Surgery Biobank and the North West Lung Centre Charity for supporting this project. In addition, we would like to thank the study participants for their contribution. S.G. was supported by the Fungal Infection Trust and she is currently funded by the NC3Rs grant: NC/P002390/1. P.B. was supported by MRC grant MR/M02010X/1. N.L.D.O. received support from the Fungal Infection Trust and the National Institute for Health Research South Manchester Respiratory and Allergy Clinical Research Facility at University Hospital of South Manchester NHS Foundation Trust. N.B.G. is supported by the Libyan Ministry of Higher Education and Scientific Research.

## Author contributions

S.G. and P.B. designed and performed the experiments, analysed data and prepared the manuscript. N.O. provided technical support to S.G., N.B.G and N.R. performed the confocal image analysis and processing of the epithelial monolayers, L.N.F. analysed *Aspergillus* fungal loads in respiratory samples from patients at the National Aspergillosis Center, helped with patient selection for genotyping studies and preparation of the manuscript, D.W.D. and P.B. conceived the project, designed and supervised this study and prepared the manuscript.

## Additional information

**Competing interests:** N.B.G., L.N.F., D.W.D. and P.B. declare the following competing interests related to this research. N.B.G. has been paid for talks on behalf of Nawgen SL. L.N.F. has been paid travel expenses and she has written and presented on resistance monitoring using pyrosequencing for Gilead GAIN education programme. D.W.D. and family hold founder shares in F2G Ltd., an antifungal discovery company and in Novocyt, which markets the Myconostica real-time molecular assays. D.W.D. has a patent for fungal infection assays that has been externally licensed; he acts or has recently acted as a consultant to Astellas, Sigma Tau, Basilea, Scynexis, Cidara, Biosergen, Quintilles, and Pulmocide; and in the last 3 years he has been paid for talks on behalf of Astellas, Dynamiker, Gilead, Merck, and Pfizer. P.B. is a founder of Alergenetica SL. and Syngenics Ltd. None of the other authors have a relevant financial conflict of interest. S. G. is a council member of the International Society of Human and Medical Mycology (ISHAM). N.B.G. has been the head of the Libyan Medical Laboratory Association. N.R. has been a President of the British Mycological Society, a Fellow of the Royal Society of Biology and of the Royal Microscopical Society. D.W.D. is a longstanding member of the Infectious Disease Society of America Aspergillosis Guidelines Group, the European Society for Clinical Microbiology and Infectious Diseases Aspergillosis Guidelines Group and the British Society for Medical Mycology Standards of Care Committee. None of the other authors have a relevant non-financial conflict of interest.

