## [Peer Review File · Nature Communications]

Reviewers' comments:

Reviewer #1 (Remarks to the Author):

The authors examine the role of the ZNF77 transcription factor and the rs35699176 variant leading to the premature termination of the protein in fungal colonization of patients with allergic bronchopulmonary aspergillosis (ABPA). The use of CRISP/Cas9-edited bronchial epithelial cells and patient samples reveals the inability of cells with the rs35699176 variant to control fungal adhesion and colonization. Although this report describes a novel role for ZNF77 during ABPA and several strategies have been developed to understand its role in fungal colonization (e.g., RNA-seq, secretome analysis, etc.), there is no concrete mechanistic insight into how functional deficiency of ZNF77 increases susceptibility to fungal colonization in ABPA. It remains unclear whether adhesion properties, secretion of soluble factors, fungal killing or even cytokine responses are in play. In addition, data on 16HBE cells lacks confirmation in primary human cells from patients. I have outlined below the major points which require further attention.

Introduction: It is not clear to me why the authors exclude a role for genetic variants in immunomodulatory genes based on the familial occurrence of ABPA. I am certain that there are variants with similar frequencies to rs35699176 that affect immune-related genes or pathways, and that may be relevant in ABPA.

A transcription factor such as ZNF77 may have multiple targets and the rs35699176 variant may influence severely the interaction of ZNF77 with its gene targets. Have the authors attempted to search binding sites for ZNF77 in the genes they have identified in the RNA-seq analysis?

Because previous reports have demonstrated a role for this variant in cytokine production, it would be useful to understand whether cytokine or chemokine production by epithelial cells (e.g., IL-8) is also affected in this context, leading to enhanced fungal colonization in ABPA.

Figure 1: The authors show that the genome editing of 16HBE cells does not influence the expression of ZNF77. However, no evidence is provided on what are the consequences of rs35699176 to the ability of ZNF77 in regulating gene expression. Did the authors check whether ZNF77 acts as a trans-eQTL affecting the expression of other genes at a distance? Also, although it does not appear to behave as a cis-eQTL, it would be important to discard its role as a context-specific eQTL measurable only during fungal infection.

The general findings of defective cellular confluence are in contrast with the increased expression of occludin detected in genome-edited cells, and no explanation is provided for this. Could the lack of confluence be attributed to differences in cellular viability following culture?

Figure 3: I have some doubts about how RNA-seq and secretome analysis are presented. There is no indication whether these analyses were performed in basal (unstimulated) conditions) or after 6h (?) infection with *Aspergillus fumigatus*. This is also not clear from the Methods section.

Increased fungal colonization could in theory also be due to an impaired ability to cells to phagocytose and clear the fungus, favoring its persistence within the host. Have the authors checked whether phagocytosis efficiency is maintained in genome-edited cells? The list of genes identified by transcriptomics is suggestive of effects on such processes.

Figure 4: The authors indicate that carriage of the heterozygous genotypes includes GA and GG. This is obviously inaccurate and should be corrected. I assume that in the figure, the authors are comparing GG vs. GA (since AA are rare) and not GG vs. GG+GA.

Is it possible to name the soluble factor (or factors) that are secreted and influence germination based the secretome data?

Reviewer #2 (Remarks to the Author):

This is a well designed study to evaluate an important mechanistic question of why some, and only some, patients with disease of impaired airway clearance (asthma and CF) develop fungal colonisation with *Asp fumigatus*. The authors focus on the gene variant (rs35699176) in the transcription factor ZNF77 in mediating disease, although it is not clear how they chose this target (where is their "Supplemental methods: Note on variant selection"?). Nonetheless, this study is a comprehensive evaluation of the impact of altered tf genetics on (a) gene expression and proteomics involving immune responses and molecular adhesion, and (b) microbiology patterns. As a follow up to this in vitro model (transfected vs non-transfected bronchial epithelial cells), they then analysed the relationship between genotype and *Aspergillus* loads in patients with asthma. A significant association with ABPA remains to be established. If that happens, the results will be more clinically relevant.

The major claim is a novel one: risk for aspergillosis in patients with chronic bronchial disease is mediated, in part, by genes which control epithelial cell function. Data is presented for adhesion molecules and immune response programs for this risk. The results will be of at least moderate interest to those in related fields, although likely less so for the wider community of medical research. These findings, if replicated, will certainly influence thinking in the field, although the direct link to clinical practice is left unclear. The claim that these findings will be of diagnostic relevance is not well founded, and it is not clear how therapy will be guided by the results. ("Development of new therapies to prevent *A. fumigatus* adherence and colonization in ABPA might be a major step for patients with the rs35699176 genotype": this statement could apply to all patients with ABPA, and not just those with the stated genotype).

The overall layout/flow/organization is adequate, but the writing is not top-notch (and not at the level of this journal).

Minor comments per section:

Methods:

- The genotype of the SNP rs35699176 was not stated explicitly
- "For validation, rs35699176 genotyping and *Aspergillus* load by means of *Aspergillus*-specific qPCR were performed in sputum from ABPA patients". Typically, validation would be done in a second cohort. Genotype results shouldn't be different from BAL vs sputum samples.
- Other experts may be better qualified to evaluate the methods of gene ontology analysis

Rather than expanding on the meaning of results, the discussion in some parts repeats the presentation of results.

Finally, the figures are good but some could be better. Fig 2 is busy: I suggest combing the data for all 6 strains (which show similar results for germination time and hyphal growth). Speaking of, the reason for testing 6 strains is not made clear. Fig 3A is nice. For 3B and 3D, did they mean fold change and fold enrichment as different concepts? Also, the meaning of fold enrichment results (3D) in the normal gene example is unclear.

Reviewer #3 (Remarks to the Author):

This manuscript describes investigations of the mechanism by which the rs35699176 polymorphism in the ZNF77 transcription factor potentially confers susceptibility to ABPA. While the data are interesting, the manuscript has some weaknesses that limit the strength of the

conclusions. Specific comments:

1) Introduction. It is never stated in the manuscript why the authors initially decided to study the rs35699176 polymorphism. Was this polymorphism identified in a previous study? What sort of data led to their hypothesis that this polymorphism might be associated with susceptibility to ABPA? The lack of this information significantly weakens the paper.

2) Results. When comparing the effects of the induced rs35699176 polymorphism in the 16HBE cells versus the presence of this polymorphism in humans, there is a critical difference. In the 16HBE cells, the polymorphism is homozygous, whereas in patients it is heterozygous. As the authors mention on line 314, it is possible that the homozygous rs35699176 polymorphism is lethal. On this basis, the phenotype of the homozygous rs35699176 polymorphism in 16HBE cells is likely to be highly exaggerated compared to heterozygous humans. Thus, it would greatly strengthen the manuscript if the authors would create a heterozygous rs35699176 16HBE cell line and determine its susceptibility to *Aspergillus* infection. Note that modified CRISPR-Cas9 plasmids that cut only a single strand of DNA already exist.

3) Fig. 1D. It appears that caveolin mRNA levels were down-regulated in the 16HBE- rs35699176 cells. Is this significant? Also, there is an error in the legend (lines 108-109) about the time points. Conceptually, the increase in occludin mRNA seems opposite of what was expected. If the cell-cell junctions are weakened by the rs35699176 polymorphism, shouldn't the mRNA levels of junctional proteins be down-regulated?

4) Fig. 2D. The data in these panels are very difficult to see. The scale of the y-axis should be reduced to 20% to make the results easier to interpret.

5) Fig. 3. The interpretation of the transcriptomic and proteomic data is very general because they focus on broad categories of genes and proteins. Although some specific proteins are listed in panel C, there is no clear linkage between these factors and the observed increase in susceptibility to *Aspergillus*. For example, it is never stated whether the rs35699176 polymorphism reduced the expression of any proteins with antimicrobial properties that might explain why culture supernatants of the 16HBE rs35699176 cells had less inhibitory properties than those of the control cells. Although the excel files are difficult to interpret because of the cryptic labeling, it appears to this reviewer that lactoferrin was significantly down-regulated in the rs35699176 cells. Could this explain the lack of growth inhibitory effects?

6) Table 1. More information is needed to interpret the results in this table. SAFS should be spelled out. The demographics of the patients (e.g., age, sex) with each of the various diagnoses should be given separately, not combined. Also, the number of patients in each of the diagnostic categories with the rs35699176 polymorphism should be listed. The BAL results (*A. fumigatus* genomes/ml) should be given for each group. It appears that the FEV1 and FVC are listed in terms of liters (but this is not stated), whereas the predicted FEV1 and FVC are possibly given in terms of percentage (but this is not stated). Finally, it should be made clear in the text that lines 206-209 refer just to Table 1.

7) Table 2. Were any other factors associated with a positive *Aspergillus* qPCR—for example, smoking, sex, FEV1 or FVC? If they were, then the authors should perform multivariate analysis to verify that the rs35699176 polymorphism was truly associated with a positive *Aspergillus* qPCR rather than just being a co-variate with some other predictor. Overall, the manuscript would benefit from the input from a statistician.

Reviewer 1

1. A transcription factor such as ZNF77 may have multiple targets and the rs35699176 variant may influence severely the interaction of ZNF77 with its gene targets. Have the authors attempted to search binding sites for ZNF77 in the genes they have identified in the RNA-seq analysis?

Answer and action taken: We agree with the reviewer about the usefulness of defining the binding site of ZNF77 to completely understand its mechanism of action. No DNA binding sites are known for this protein. It is unfortunately also not currently technically feasible to perform such studies as no available antibody to ZNF77 is specific to this protein, as it has several close paralogs. We have significant ChIPseq experience in our lab and attempts to identify binding sites have been unsuccessful using 5 different anti-ZNF77 antibodies including several monoclonals. The difficulty is caused by cross reactivity which we have demonstrated using western blots. Moreover, there are no available chipseq studies in online databases that we could use to identify ZNF77 binding sites – probably for the same reason. We currently regard such studies as technically unfeasible and therefore outside the scope of this work.

2. Because previous reports have demonstrated a role for this variant in cytokine production, it would be useful to understand whether cytokine or chemokine production by epithelial cells (e.g., IL-8) is also affected in this context, leading to enhanced fungal colonization in ABPA.

Answer and action taken: The reviewer is right. We have now quantified cytokine production using the Bioplex inflammation panel from Biolegend. We have found that at 6 h post infection 16HBE^{rs35699176} cells secrete IL-10 and IL-6 in response to A.

fumigatus. We couldn't detect IL-8 in our challenge experiments and this might be due to the fact that IL-8 release correlates with damage of the lung epithelium at later stages of infection and there is no epithelial disruption at 6 h post infection. These results have been incorporated into the new version of the manuscript (L167/L172) and supplemental methods.

3. Figure 1: The authors show that the genome editing of 16HBE cells does not influence the expression of ZNF77. However, no evidence is provided on what are the consequences of rs35699176 to the ability of ZNF77 in regulating gene expression. Did the authors check whether ZNF77 acts as a trans-eQTL affecting the expression of other genes at a distance? Also, although it does not appear to behave as a cis-eQTL, it would be important to discard its role as a context-specific eQTL measurable only during fungal infection.

Answer and action taken: We note that rs35699176 introduces a stop codon into the gene so that a truncated protein with no DNA binding region is produced. It is therefore very unlikely that the variant produces active ZNF77 (L72/L78). We agree that as a transcription factor the wild-type ZNF77 protein should change expression of genes at trans loci. Clearly it is not possible to infer QTL type characteristics using only a single monoclonal cell line and a single monoclonal variant line. We have searched Trans-eQTL databases to investigate the consequences of rs35699176 in regulating gene expression however we are unable to show any such activity from the available online data, probably as this is a largely unstudied transcription factor. Again this may change as these databases and inference tools expand and improve but it is not currently feasible to infer eQTL characteristics for this gene in this context.

4. The general findings of defective cellular confluence are in contrast with the increased expression of occludin detected in genome-edited cells, and no explanation is provided for this. Could the lack of confluence be attributed to differences in cellular viability following culture?

Answer and action taken: The reviewer is right to point this out. In the new version of the manuscript, we have suggested that 16HBE^{rs35699176} cells increase occludin expression to counterbalance the downregulation of other genes and proteins involved in cell-cell adhesion and tight junction formation as detected by RNA-seq. Trypan blue staining following culture was initially performed to investigate if the lack of confluence during TEER experiments were due to differences in cell viability, however the percentage of dead cells was below 1% for both 16HBE and 16HBE^{rs35699176} (L302/L308).

5. Figure 3: I have some doubts about how RNA-seq and secretome analysis are presented. There is no indication whether these analyses were performed in basal (unstimulated) conditions) or after 6h (?) infection with *Aspergillus fumigatus*. This is also not clear from the Methods section.

Answer and action taken: We agree with the reviewer and this has now been clarified in the current version of the manuscript. Secretome analysis was performed in cell culture supernatants from 16HBE and 16HBE^{rs35699176} cells to identify basal differences between cell lines. Secretome analysis in *A. fumigatus* challenge experiments has not been included in the manuscript as at 6h post infection with the low inoculum no statistically significant differences were detected (L196/L218).

RNA-seq experiments were performed in 16HBE and 16HBE^{rs35699176} cells challenged with *A. fumigatus* spores and unchallenged controls at 6 h post infection. Comparison were performed as follows: 16HBE vs 16HBE^{rs35699176}, 16HBE + *A. fumigatus* vs 16HBE^{rs35699176} + *A. fumigatus*, 16HBE vs 16HBE + *A. fumigatus* and 16HBE^{rs35699176} vs 16HBE^{rs35699176} + *A. fumigatus*. This has been now clarified in the new version of the manuscript (L219/L241).

RNA-seq data and gene ontology analysis has been simplified as presented in a new Figure 3 and included in the results section and supplemental row data..

6. Increased fungal colonization could in theory also be due to an impaired ability to cells to phagocytose and clear the fungus, favoring its persistence within the host. Have the authors checked whether phagocytosis efficiency is maintained in genome-edited cells? The list of genes identified by transcriptomics is suggestive of effects on such processes.

Answer and action taken: We thank the reviewer for this useful comment. The process of spore swelling exposes host recognisable spore surface components that allow human cells to recognise and phagocytose spores. If spores swell more rapidly this should lead to faster recognition and phagocytosis. We have now quantified differences in the percentage of internalization between 16HBE and 16HBE^{rs35699176} challenged with *A. fumigatus* spores for 2, 4 and 6 hours. Although the percentage of internalization was slightly but significantly higher at 2 h post – challenge in the variant cell line, the internalization rate and number of internalised spores was identical to 16HBE at 4 and 6h post infection. This correlates with the spore germination data where spores swell (0-3h) and then germinate (3-6h) more rapidly in the presence of 16HBE^{rs35699176} cells. Additionally spores adhere more rapidly to the variant cell line which should also promote earlier phagocytosis. Spore phagocytosis is not an efficient killing mechanism for *A. fumigatus* and so earlier phagocytosis may actually protect spores against effective clearance mechanisms such as macrophage and neutrophil attack and mucociliary clearance. These results are included in the new version of the manuscript (Figure2, L158/L166, L318/L330 and supplemental information).

7. Figure 4: The authors indicate that carriage of the heterozygous genotypes includes GA and GG. This is obviously inaccurate and should be corrected. I assume that in the figure, the authors are comparing GG vs. GA (since AA are rare) and not GG vs. GG+GA.

Answer and action taken: We thank the reviewer for this observation. Figure 4 has been now modified.

8. Is it possible to name the soluble factor (or factors) that are secreted and influence germination based the secretome data?

Answer and action taken: We thank the reviewer for this comment. This is not possible given the very high complexity of the combined secretome and metabolome of the infection system. We did notice that lactoferrin and transactoferrin were downregulated in 16HBE^{rs35699176} cells as measured by RNA-seq and secretome analysis respectively. We previously attempted to identify the role that iron sequestration might have in our model by using 0.2mM of the ferrous iron chelator bathophenanthroline disulfonic acid. However, those experiments were not conclusive and no differences in germination or cell damage were found.

9. Reviewer 1 indicates that the manuscript would benefit from validation on primary cells carrying the rs35699176 genotype.

Answer and action taken: We agree with the reviewer and validation using primary cells will be very useful. However, patients with asthma or the fungal diseases discussed in this paper do not usually undergo biopsy for clinical reasons, and on the very rare occasions where bronchoscopy mediated biopsy is performed the level of fungus renders isolation of epithelial cells nearly impossible – in our hands < 3% of biopsies from fungal patients produce usable epithelial cells - The individuals carrying the mutation are demonstrably those with the highest levels of fungus in their lungs therefore we decided to validate our results directly with clinical phenotypes as shown in the exploratory cohort in this study..

Reviewer 2:

1. The claim that these findings will be of diagnostic relevance is not well founded, and it is not clear how therapy will be guided by the results. ("Development of new therapies to prevent *A. fumigatus* adherence and colonization in ABPA might be a major step for patients with the rs35699176 genotype": this statement could apply to all patients with ABPA, and not just those with the stated genotype).

Answer and action taken: We dispute the reviewers comment. Fungal diseases are extremely difficult to diagnose and the observation that individuals carrying this mutation have 17-fold higher levels of fungus in the lung is of clear relevance in the context of disease surveillance and therapy. Higher levels of fungus certainly mean higher levels of allergen and probably higher levels of tissue damage leading to inflammatory responses. Therefore use of this marker to define individuals in at risk populations has clear diagnostic potential. We agree with the reviewer that the sentence is unclear and we have clarified this statement in the new version of the manuscript.

2. The overall layout/flow/organization is adequate, but the writing is not top-notch (and not at the level of this journal).

Answer and action taken: The paper has been extensively rewritten and the new version of the manuscript has been carefully edited using as template recently accepted articles in Nature Communications in this field.

3. Minor comments per section:

- The genotype of the SNP rs35699176 was not stated explicitly: **Answer and action taken:** We agree with the reviewer and this has been clarified in the new version of the manuscript.
- "For validation, rs35699176 genotyping and Aspergillus load by means of Aspergillus-specific qPCR were performed in sputum from ABPA patients". Typically, validation would be done in a second cohort. Genotype results shouldn't be different from BAL vs sputum samples. **Answer and action taken:** We agree with the reviewer and we have now clarified this point in the new version of the manuscript. We decided to perform validation using sputum samples as they are the standard sample using to estimate fungal colonization in fungal allergy. The genotype results are different because this is not a disease genotyping study - fungal level was chosen as the primary marker for association with the allele and not disease.(L256/L270, Figure 4, Table 1, Table 2 and Methods).
- Other experts may be better qualified to evaluate the methods of gene ontology analysis. **Answer and action taken:** Thanks for this comment.
- Rather than expanding on the meaning of results, the discussion in some parts repeats the presentation of results. **Answer and action taken:** We agree with the reviewer and the discussion section has been rewritten in the new version of the manuscript (L279/L374).
- Finally, the figures are good but some could be better. Fig 2 is busy: I suggest combing the data for all 6 strains (which show similar results for germination time and hyphal growth). Speaking of, the reason for testing 6 strains is not made clear. Fig 3A is nice. For 3B and 3D, did they mean fold change and fold enrichment as different concepts? Also, the meaning of fold enrichment results (3D) in the normal gene example is unclear. **Answer and action taken:** As per the reviewers suggestion figures 2 and 3 have been simplified in the new version of the manuscript.

Reviewer 3

1. Introduction. It is never stated in the manuscript why the authors initially decided to study the rs35699176 polymorphism. Was this polymorphism identified in a previous study? What sort of data led to their hypothesis that this polymorphism might be associated with susceptibility to ABPA?The lack of this information significantly weakens the paper.

Answer and action taken: We appreciate the reviewers comments and we have expanded the explanation about why we chose ZNF77 as target for our analysis. We did discuss this with the editor prior to submission to clarify this point (L68/L78).

2. Results. When comparing the effects of the induced rs35699176 polymorphism in the 16HBE cells versus the presence of this polymorphism in humans, there is a critical difference. In the 16HBE cells, the polymorphism is homozygous, whereas in patients it is heterozygous. As the authors mention on line 314, it is possible that the homozygous rs35699176 polymorphism is lethal. On this basis, the phenotype of the homozygous rs35699176 polymorphism in 16HBE cells is likely to be highly exaggerated compared to heterozygous humans. Thus, it would greatly strengthen the manuscript if the authors would create a heterozygous rs35699176 16HBE cell line and determine its susceptibility to *Aspergillus* infection. Note that modified CRISPR-Cas9 plasmids that cut only a single strand of DNA already exist.

Answer and action taken: This is a useful comment. Our cell culture model can only represent very early events in infection and is used only as an indicative model – rather than as a surrogate model for pathogenicity. In this context we don't think that heterozygous mutant lines will add much – most likely reduced phenotypes of the types already shown. The infection model is limited and does not give data on *Aspergillus* infection directly so little additional information would be generated using a heterozygous system. We would therefore argue that the benefit of creating such a cell line is limited. We note that this tool was not available at the time we started the study (3 years ago) and the recreation of the heterozygous variant together with validation will take at least 2 more years. Moreover, as we could see a clear phenotype associated with our in vitro results in patients carrying the heterozygous rs35699176 variant we feel confident in our observations. We will certainly consider this approach for future experiments but would still most likely start by constructing the homozygous mutation in order to provide the cleanest phenotype.

3. Fig. 1D. It appears that caveolin mRNA levels were down-regulated in the 16HBE-rs35699176 cells. Is this significant? Also, there is an error in the legend (lines 108-109) about the time points. Conceptually, the increase in occludin mRNA seems opposite of what was expected. If the cell-cell junctions are weakened by the rs35699176 polymorphism, shouldn't the mRNA levels of junctional proteins be down-regulated?

Answer and action taken: We appreciate the reviewer observation. Differences in caveolin expression in the 16HBE^{rs35699176} cell line were lower than in 16HBE cells but the difference in levels was not significant. In general we feel that dysregulation of junction components rather than simple downregulation is responsible for the phenotype (see comments above). We have also clarified in the new version of the manuscript that higher occludin expression in 16HBE^{rs35699176} cells could be associated with a mechanism to compensate the down regulation of other tight junction and cell-cell adhesion proteins leading to certain confluence. This has been discussed in the new version of the manuscript.

4. Fig. 2D. The data in these panels are very difficult to see. The scale of the y-axis should be reduced to 20% to make the results easier to interpret.

Answer and action taken: Yes- good point. Axes in Figure 2 plots have been resized in the new version of the manuscript.

5. Fig. 3. The interpretation of the transcriptomic and proteomic data is very general because they focus on broad categories of genes and proteins. Although some specific proteins are listed in panel C, there is no clear linkage between these factors and the observed increase in susceptibility to *Aspergillus*. For example, it is never stated whether the rs35699176 polymorphism reduced the expression of any proteins with antimicrobial properties that might explain why culture supernatants of the 16HBE rs35699176 cells had less inhibitory properties than those of the control cells. Although the excel files are difficult to interpret because of the cryptic labeling, it appears to this reviewer that lactoferrin was significantly down-regulated in the rs35699176 cells. Could this explain the lack of growth inhibitory effects?

Answer and action taken: The reviewer is right and the RNA-seq and Secretome Gene Ontology data has been simplified in the new version of the manuscript. As mentioned to reviewer 1, lactoferrin and transactoferrin were downregulated in 16HBE^{rs35699176} cells as measured by RNA-seq and secretome analysis respectively. We previously attempted to identify the role that iron sequestration might have in our model by using 0.2 mM of the ferrous iron chelator bathophenanthroline disulfonic acid. However, a significant effect of iron or chelation of iron in those experiments was not observed. Only a limited number of antifungal proteins and metabolite are known and these were not significantly regulated by the mutation.

6. Table 1. More information is needed to interpret the results in this table. SAFS should be spelled out. The demographics of the patients (e.g., age, sex) with each of the various diagnoses should be given separately, not combined. Also, the number of patients in each of the diagnostic categories with the rs35699176 polymorphism should be listed. The BAL results (*A. fumigatus* genomes/ml) should be given for each group. It appears that the FEV1 and FVC are listed in terms of liters (but this is not stated), whereas the predicted FEV1 and FVC are possibly given in terms of percentage (but this is not stated). Finally, it should be made clear in the text that lines 206-209 refer just to Table 1.

Answer and action taken: The reviewer is right - this is clarified in the new version of the manuscript.

7. Table 2. Were any other factors associated with a positive *Aspergillus* qPCR —for example, smoking, sex, FEV1 or FVC? If they were, then the authors should perform multivariate analysis to verify that the rs35699176 polymorphism was truly associated with a positive *Aspergillus* qPCR rather than just being a co-variate with some other predictor. Overall, the manuscript would benefit from the input from a statistician

Answer and action taken: This is a useful comment and we agree that the data, as presented, is not clear. Table two has been updated in the new version of the manuscript. We did not find any association between lung function or any other

demographic variant and the rs35699176 genotype except fungal burden as stated. Results were tested using logistic regression and by stratifying patients using the various conditions mentioned, where feasible, and no further or confounding associations were observed. We have revised the statistical analyses in the current version of the manuscript (Table 1 and Table 2).

Reviewers' comments:

Reviewer #1 (Remarks to the Author):

The authors have adequately addressed most of the issues that were raised previously, and now provide additional data strengthening the manuscript. I have a few additional comments:

1. The finding that the genome-edited cells produce higher amounts of IL-10 after infection is a relevant finding. Recent work by Cunha and colleagues (J Allergy Clin Immunol, 2017) demonstrated that genetic variants leading to enhanced IL-10 production after infection were associated with risk for invasive aspergillosis and compromised the ability of macrophages in clearing *Aspergillus*. This important piece of data should be addressed in the discussion.
2. Please provide the numbers of patients that were genotyped in the original and replication cohorts in supplementary table 1. Was the same genotyping method used for both cohorts?

Reviewer #2 (Remarks to the Author):

The authors have addressed many of the concerns raised on the initial review. The revised manuscript is better for it. The figures in particular are much improved.

"The claim that these findings will be of diagnostic relevance is not well founded, and it is not clear how therapy will be guided by the results." This comment is not well refuted by the authors, who said "We dispute the reviewers comment. Fungal diseases are extremely difficult to diagnose and the observation that individuals carrying this mutation have 17-fold higher levels of fungus in the lung is of clear relevance in the context of disease surveillance and therapy." It is not clear how their findings will help in the diagnosis of fungal disease. The results could help identify patients at risk for disease, but that is different than establishing a diagnostic test. The fact that fungal diseases are "extremely hard" to diagnose does not make their genetic findings a legitimate diagnostic platform. Testing the results in clinical studies which demonstrate the utility of genotyping to confirm or refute a diagnosis will do that.

The reference to fibromyalgia is clinically irrelevant to pulmonary disease; it remains unclear how they identified which polymorphisms to target for study.

Reviewer #3 (Remarks to the Author):

The revised manuscript is much improved and satisfactorily addresses all of the comments.

Authors' Response to Reviewers

Reviewer #1 (Remarks to the Author):

The authors have adequately addressed most of the issues that were raised previously, and now provide additional data strengthening the manuscript. I have a few additional comments:

1. The finding that the genome-edited cells produce higher amounts of IL-10 after infection is a relevant finding. Recent work by Cunha and colleagues (J Allergy Clin Immunol, 2017) demonstrated that genetic variants leading to enhanced IL-10 production after infection were associated with risk for invasive aspergillosis and compromised the ability of macrophages in clearing *Aspergillus*. This important piece of data should be addressed in the discussion.

Answer and action taken: We thank the reviewer for this question and this has now been included in the new version of the manuscript as follows “It has recently described that a genetically conditioned overexpression of IL-10 predisposes to invasive aspergillosis by suppressing the anti-*Aspergillus* response of macrophages Cunha C et al (J Allergy Clin Immunol, 2017)”

1. Please provide the numbers of patients that were genotyped in the original and replication cohorts in supplementary table 1. Was the same genotyping method used for both cohorts?

Answer and action taken: We agree the reviewer's comment and we have clarified this in Supplemental Table 1 and in the discovery supplemental note. DNA from 96 subjects with ABPA and 167 asthmatic controls was exome sequenced at the Centro Nacional de Analisis Genomico (CNAG) according to Overton NLD et al (Plos One, 2018). For replication, DNA from 96 ABPA and 96 asthmatics not originally included in the discovery cohort was sequenced by MiSeq. (Overton NLD, Brakhage AA,

Thywißen A, Denning DW, Bowyer P (2018) Mutations in EEA1 are associated with allergic bronchopulmonary aspergillosis and affect phagocytosis of *Aspergillus fumigatus* by human macrophages. PLoS ONE 13(3): e0185706. <https://doi.org/10.1371/journal.pone.0185706>

Reviewer #2 (Remarks to the Author):

The authors have addressed many of the concerns raised on the initial review. The revised manuscript is better for it. The figures in particular are much improved.

1. "The claim that these findings will be of diagnostic relevance is not well founded, and it is not clear how therapy will be guided by the results." This comment is not well refuted by the authors, who said "We dispute the reviewers comment. Fungal diseases are extremely difficult to diagnose and the observation that individuals carrying this mutation have 17-fold higher levels of fungus in the lung is of clear relevance in the context of disease surveillance and therapy." It is not clear how their findings will help in the diagnosis of fungal disease. The results could help identify patients at risk for disease, but that is different than establishing a diagnostic test. The fact that fungal diseases are "extremely hard" to diagnose does not make their genetic findings a legitimate diagnostic platform. Testing the results in clinical studies which demonstrate the utility of genotyping to confirm or refute a diagnosis will do that.

Answer and action taken: The reviewer is right and we have changed our statement at the end of the introduction as "Based on these results, we propose that ZNF77-genotyping of patients with ABPA may be a useful risk-marker for fungal colonization."

2. The reference to fibromyalgia is clinically irrelevant to pulmonary disease; it remains unclear how they identified which polymorphisms to target for study.

Answer and action taken: The reviewer is right and the references about fibromyalgia have been deleted from the manuscript. Moreover, we have included some clarification about how this polymorphism was discovered and included the reference of the methodology of whole exome sequencing study done in our group (Overton NLD, Brakhage AA, Thywißen A, Denning DW, Bowyer P (2018) Mutations in EEA1 are associated with allergic bronchopulmonary aspergillosis and affect phagocytosis of *Aspergillus fumigatus* by human macrophages. PLoS ONE 13(3): e0185706. <https://doi.org/10.1371/journal.pone.0185706>)

Reviewer #3 (Remarks to the Author):

The revised manuscript is much improved and satisfactorily addresses all of the comments.

Answer and action taken: We thank the reviewer for this positive comment.

REVIEWERS' COMMENTS:

Reviewer #1 (Remarks to the Author):

The authors have addressed my comments on the revised manuscript, and I have no new concerns with this version. The revised manuscript is very clear and makes the points well. The authors are to be commended for their efforts in the revision.

Reviewer #1 (Remarks to the Author):

The authors have addressed my comments on the revised manuscript, and I have no new concerns with this version. The revised manuscript is very clear and makes the points well. The authors are to be commended for their efforts in the revision.

- >Action taken: We thank the reviewer for his/her contribution in reviewing the manuscript.